# Synergistic Effects of Waste Glass Powder, High-Frequency Ultrasonic Dispersion, and Liquid Glass Treatment on the Properties of Aluminum-Based Ultra-Lightweight Concrete

**DOI:** 10.3390/ma17225430

**Published:** 2024-11-07

**Authors:** Evaldas Serelis, Vitoldas Vaitkevicius, Siavash Salehi, Maris Sinka, Alise Sapata

**Affiliations:** 1Faculty of Civil Engineering and Architecture, Kaunas University of Technology, Studentų g. 48, 51367 Kaunas, Lithuania; evaldas.serelis@ktu.lt (E.S.); siavash.salehi1992@gmail.com (S.S.); 2Institute of Materials and Structures, Riga Technical University, Kipsala Street 6A, LV 1048 Riga, Latvia; maris.sinka@rtu.lv (M.S.); alise.sapata@rtu.lv (A.S.)

**Keywords:** waste glass powder, high-frequency ultrasonic dispersion, liquid glass, ultra-lightweight concrete, compressive strength

## Abstract

This research investigates the impact of waste glass powder, high-frequency ultrasonics (HFUS) dispersion, and liquid glass treatment on aluminum-based ultra-lightweight concrete. Substituting up to 80% of Portland cement with waste glass powder significantly delays hydration and reduces compressive strength by 77%. However, applying HFUS dispersion for 60 s to a mixture with 30% waste glass powder substitution restored compressive strength to the reference value of 3.1 MPa. The combined HFUS and liquid glass treatment enhanced compressive strength by 87%, increased density by 32%, and significantly reduced prosody. Scanning electron microscopy revealed a progressively denser cement matrix with each treatment, highlighting the synergistic effects of these methods in improving concrete properties.

## 1. Introduction

Over recent decades, advances in waste processing technologies have significantly reduced landfilling while promoting alternative waste treatment methods. Waste can be categorized into various types, including construction and demolition waste, radioactive waste, energetic waste, hazardous waste, electronic waste, mixed waste, medical waste, and municipal waste [1,2,3]. Municipal solid waste comprises household trash such as packaging, grass clippings, furniture, clothing, bottles, food scraps, newspapers, appliances, paint, and batteries. It also includes dirt, sand, stones, metals, wood, glass, mirrors, dishes, and plastic [4,5,6]. Waste from industrial, hospital, or radioactive facilities is excluded from this category [7,8].

Europe shows a positive trend in recycling, although disparities exist between countries. Eurostat reports a decline in landfilled waste despite increased overall waste generation in the EU. I Wayan Koko Suryawan’s research highlights a direct correlation between landfilled municipal waste and greenhouse gas emissions [9]. The EU saw a 55% reduction in landfilled municipal waste, from 121 million tons in 1995 to 54 million tons in 2021, with an annual decrease of about 3.0%. This reduction supports the EU’s goal to promote a circular economy by converting waste into secondary raw materials or new products [10]. As a result, the interest in recycling is expected to grow.

A. Karagiannidis suggested categorizing municipal waste into solid, semi-solid, liquid, and gaseous forms to facilitate processing via incineration, gasification, or pyrolysis [11]. Eurostat data show that waste incineration in the EU increased by 107%, from 30 million tons in 1995 to 62 million tons in 2021. Recycling for secondary raw materials grew by 203%, from 23 million tons to 70 million tons, and composting rose by 218%, from 14 million tons to 45 million tons. These technologies are widely used globally but have drawbacks. Hongtao Sun pointed out that incineration generates bottom ash and fly ash, which contain high levels of heavy metals such as Cu, Pb, Zn, and Cd, limiting its use [12]. Ahmad Assi’s research indicates that incineration produces about 75–90% bottom ash and up to 5% fly ash [13]. Managing these byproducts is challenging, as new landfills may become necessary without proper measures.

Recycling municipal waste into secondary raw materials is not always efficient. Specialized containers help but often contain impurities. Glass containers, for instance, may include metals, wood, and batteries, while plastic containers might hold food scraps, paper, and metal. This contamination issue varies across the EU, with a gap between Eastern and Western Europe. To address this, many countries invest in advanced waste sorting systems to extract valuable materials.

Recycling one ton of flat glass from construction saves up to 1200 kg of raw materials and 25% of energy, reducing CO_2_ emissions by about 300 kg [14,15]. Recycling glass wool and mineral wool offers similar energy savings and CO_2_ reductions [16,17]. Using recycled materials in gypsum boards can reduce CO_2_ emissions by up to 5% when 25% of the material is recycled [18,19]. Pre-sorting waste improves incineration efficiency and conserves resources. However, incineration is only suitable for high-calorific-value waste, such as wood, paper, and plastic, while dirt and food leftovers are better suited for composting [20]. Metals like aluminum, iron, copper, steel, and brass can also be efficiently sorted and recycled [21]. Clean, sorted materials are more valuable and have broader recycling applications.

This research examines the use of waste glass shards from municipal waste sorting processes. These shards are often contaminated with paper, aluminum, dirt, stones, food scraps, wood, and clothing. Such contamination complicates their reuse of new bottles, jars, or windows. However, these glass shards can still be used in civil engineering to produce various building materials, though detailed examination is needed to address potential issues.

Glass, with its high SiO_2_ content (≥60%), is amorphous and chemically similar to Portland cement, suggesting it could exhibit cement-like properties [22,23]. In construction, glass powder typically replaces Portland cement, microfillers, and aggregates. Initially, glass powder acts as an inert filler, gaining beneficial properties over time. Researchers have shown that finely ground glass powder can function as a pozzolanic material, participating in secondary hydration processes to form low-basicity calcium hydrosilicates [24,25]. Yingbin Wang noted that these pozzolanic properties emerge when glass powder particles are smaller than 75 µm [26]. When combined with lime, Keren Zheng confirmed that glass powder could form crystalline C-(N)-S-H [27]. Mohammadreza Mirzahosseini found that glass powder in Portland cement systems is effective at higher temperatures (≥60 °C) [28]. B. Venkatesan discovered that blending glass powder in cement mortars can increase 90-day compressive strength by up to 20% [29]. Qingyu Cao’s research demonstrated that glass powder helps maintain high pH levels in pore solutions, preventing steel corrosion [30].

However, some studies indicate that glass powder has low early-stage pozzolanic reactivity. Evaldas Serelis found that high-frequency ultrasonic (HFUS) dispersion can accelerate hydration in Portland cement and silica fume, but the pozzolanic effect of glass powder is initially minimal [31]. Sihai Bao’s XRD analysis revealed stable portlandite peaks, suggesting low pozzolanic reactivity of fine glass powder [32]. Glass powder can form sodium silicate gel, similar to liquid glass, which accelerates cement hydration [33,34]. Sile Hu prepared calcium silicate hydrate (C-S-H) seed from rising hush ash, which enhanced the hydration process of Portland cement; this increased compressive strength in blended cement systems [35]. Kailun Chen suggested that recycled glass powder undergoes geopolymerization, forming a three-dimensional polymeric chain with Si-O-Al-O bonds, significantly affecting the modified ternary geopolymer gel structure [36]. Nattapong Chuewangkam demonstrated that geopolymer mixtures with fly ash achieve high compressive strength after thermal curing [37].

Despite its potential, using glass powder in construction faces regulatory and contamination challenges. EU legislation often restricts its use in building products, though exceptions exist for expanded glass, glass wool, countertops, road pavement elements, and landscaping. Contaminants like aluminum, food scraps, dirt, and wood complicate reuse. Elemental aluminum reacts with alkalis, forming hydrogen gas and increasing concrete porosity. Food and dirt contamination can cause swelling or fermentation, delaying cement hydration. D.R. Kumar found that angular glass powder decreases workability, an issue mitigated by adding fly ash and silica fume [38]. Bhargavi N. Kulkarni’s research indicated that soluble impurities cause swelling in cementitious and geopolymer concrete [39]. Alkali-silica reactions between cement paste and reactive aggregates can cause structural damage. Dip Banik found that replacing PLC with 4.75% nano-SiO_2_ can mitigate ASR, whereas nano-SiO_2_ with 67 nm showed the highest mitigation [40]. Haoliang Jin concluded that pozzolanic materials like silica fume, fly ash, or metakaolin can mitigate deleterious expansions [41]. Zine el-Abidine Kameche showed that milling glass powder to ≤75 µm can induce pozzolanic reactions instead of alkali-silica responses [42].

The authors propose using waste glass powder in aluminum-based ultra-lightweight concrete (≤800 kg/m^3^), a cost-effective alternative to autoclaved aerated concrete. This process requires minimal equipment and energy. The detrimental effects of waste glass powder were mitigated using high-frequency ultrasonic dispersion. Some samples were submerged in liquid glass and kept under a negative vacuum for 30 min. This applied technique increased microstructure and compressive strength even further. This research explores waste glass shards in ultra-lightweight aluminum-based concrete. This innovative approach has not been extensively studied and could be applied to other concrete products.

## 2. Materials

Cement. Portland cement CEM I 42.5 R (provided by the Akmenės cementas AB) was employed in this study. Its primary characteristics are as follows: normal consistency paste—28.5%, specific surface area by Blaine—3719 cm^2^/kg, soundness by Le Chatelier—1.0 mm, initial setting time—110 min, final setting time—210 min, compressive strength (2/28 days)—32.2/62.8 MPa. Mineralogical composition includes: C_3_S—68.65%, C_2_S—8.75%, C_3_A—0.25%, C_4_AF—16.15%. XRD patterns of unhydrated cement are shown in Figure 1, particle size distribution in Figure 2, and chemical composition in Table 1.

Waste Glass Powder. Waste glass shards (prepared by the authors) sourced from municipal solid waste were utilized in this research. The shards were milled to achieve a specific surface area of 3711 cm^2^/g and a specific density of 2304 kg/m^3^. The XRD patterns of unhydrated waste glass powder are presented in Figure 1, while the particle size distribution is illustrated in Figure 2. The chemical composition is detailed in Table 1. The preparation for the waste glass powder is comprehensively described in the Section 3.

Gypsum (β-calcium sulfate hemihydrate). The critical properties of the gypsum (provided by the Knauf SIA) are specific surface area by Blaine—4850 cm^2^/g, specific density—2322 kg/m^3^, and average particle size—34.42 µm. Its chemical composition is as follows: CaO—37.9%, SO_3_—51.1%, insoluble—4.1%, and ignition loss—7.1%. The particle size distribution is depicted in Figure 2.

Sodium Hydroxide (NaOH). Sodium hydroxide (provided by the UAB “Lerochemas”) granules were utilized in this study. They possess the following fundamental properties: density—2.12 g/cm^3^, water solubility—415 g/L at 0 °C, 1105 g/L at 20 °C, and 3365 at 100 °C, with a molar mass of 39.99 g/mol. The purity of the NaOH granules was ≥99%.

Aluminum Powder. The aluminum Powder (provided by “Benvita” UAB) used in the tests exhibits the following primary properties: assay—99.4 trace metal basis, form—powder, auto-ignition temperature—760 °C, resistivity—2.6502 μΩ-cm, boiling point—2461 °C, melting point—662 °C, and density—2.68 g/mL at 25 °C (lit.). The particle size distribution of the aluminum powder is depicted in Figure 2.

Polypropylene Fiber. The polypropylene fibers (provided by the Baumhueter extrusion GmbH) used in this study possess the following fundamental properties: raw material—polypropylene, shape—round, diameter—16 µm, fiber length—12 mm, fiber quantity—270 million pcs/kg, flexibility modulus—3500–3900 N/mm^2^, flammable temperature ≥ 320 °C, electrical conductivity—0, and chemical resistance—good.

## 3. Methods

Waste Glass Preparation. Municipal solid waste was obtained from a local sorting plant. Aged waste was screened, shredded, and processed through a wind separator. Light combustible materials were sent to incineration, while metals and heavy inert materials were separated. Glass shards (2–6 mm) collected by the wind separator were milled in a ball mill (drum diameter—70 cm, length—52 cm, ball diameter—6.35 cm, ball mass—1.03 kg, 120 balls per batch). In each milling attempt, 4.00 kg of glass shards were processed. The resulting glass powder was sieved through a 1.00 mm sieve to remove impurities. The conversion process of waste glass shards into waste glass powder is illustrated in Figure 3.

Specific Surface Area and Particle Size Distribution. The surface area was determined using a Blaine instrument according to the EN 196-6 standard [43]. The particle size distribution was analyzed using a MasterSizer instrument (Malvern Instruments Ltd., Worcestershire, UK).

Liquid Glass Preparation. Waste glass powder, characterized by a specific surface area of 3711 cm^2^/g and a particle size distribution ranging from 0.55 µm to 69.18 µm, was utilized to produce liquid glass. Initially, 400 g of waste glass powder was added to a stainless-steel container containing 1000 mL of 5 M NaOH solution. The mixture was placed on a magnetic stirrer with the following specifications: capacity—10 L, speed range—0–1400 rpm, temperature control range 5–400 °C, heating power—1000 W, stirring power—40 W. The solution was stirred at 80 °C for 24 h to facilitate dissolution.

Mixing, Sample Preparation, and Curing. Concrete mixes were prepared using a high-intensity drill mixer with specifications: a power input of 1300 Watts, a no-load speed ranging from 0 to 900 rpm, a weight of 5.2 kg, and a mixing capacity of up to 50 kg. 

Before mixing, all ingredients were individually weighed. A foaming agent consisting of sodium hydroxide and aluminum powder was prepared separately. Sodium hydroxide dissolved aluminum powder, generating hydrogen gas as a byproduct and causing the mixture to expand. Subsequently, gypsum, acting as a structure-forming agent, was added, promoting the growth of ettringite crystal for early strength development in the plastic state. 

Sodium hydroxide was dissolved in separate containers, and a portion of water was deducted from the total water content used in slurry preparation. The water was then allowed to dissolve for 10 min. Dry gypsum and aluminum powder were added to the mix as powder constituents. The detailed mixing procedure is outlined in Table 2.

The prepared mixture exhibited a slurry consistency similar to self-compacting concrete, facilitating seamless pouring into molds (Figure 4). Samples were left to cure for 24 h before demolding. Some samples were subjected to curing for 28 days at 20 °C and 40–60% RH; some samples, after demolding, were submerged into prepared liquid glass and allowed under negative vacuum (−100 bar) to soak for 30 min; after that, kept with the rest of the samples. The dimensions of all samples cast for the experiments were 100 × 100 × 100 mm. As no additional sand or coarse aggregates were used in the concrete samples, the same crushed specimens were prepared for other analyses, including X-ray Diffraction (XRD) and Scanning Electron Microscopy (SEM). The notations for the binder used in the XRD analysis and the prepared concrete samples are consolidated in Table 3. Different compositions of aluminum-based ultra-lightweight concrete analyzed in the research are shown in Table 4.

High-Frequency Ultrasonic Dispersion (HFUS). The binder hydration process was intensified using a high-frequency ultrasonic dispersion apparatus, the “SONOPULS HD 3400”, manufactured by BALDELIN Electronic GmbH and Co. (Berlin, Germany). The key specifications of the device include an operating volume capacity of up to 2500 mL, a high-frequency generator (GM 3400), an ultrasonic converter (UW 3400), and a booster horn (SH 3425).

X-ray Diffraction Analysis (XRD). X-ray diffraction analysis was conducted on the hardened cement pastes using an XRD 3003 TT diffractometer (GE Sensing and Inspection Technologies GmbH, Hürth, Germany) with θ−θ configuration and CuKα radiation (λ = 1.54 Å). Measurements were taken over an angular range of 5° up to 70° 2 Theta, with a step width of 0.02° and a measuring time of 6 s per step. For quantitative phase analysis via the Rietveld refinement method, samples were mixed with 20 wt.% ZnO, a commonly used internal standard in XRD analysis, is stored in an argon atmosphere until measurement. The non-crystalline phase content was determined using the Rietveld fitting procedure. 

Mercury Intrusion Porosimetry (MIP) Analysis. At 28 days, cubic specimens (100 × 100 × 100 mm) from each composition were fractured into small fragments and immersed in isopropanol to stop hydration. The fragments were then dried at 40 °C to remove residual free water and stored in sealed containers for MIP analysis. During the MIP procedure, pressure was incrementally increased from 0 MPa to 450 MPa. A constant contact angle of 140° and mercury surface tension of 480 mN/m were applied to calculate pore size distribution.

Scanning Electron Microscopy (SEM) and Energy-Dispersive X-ray Spectroscopy (EDS) Analysis. The microstructure of the hardened ultra-lightweight concrete (ULWC) samples was examined using a high-resolution scanning electron microscope, specifically the FEI Quanta 200 FEG equipped with a Schottky field emission gun (FEG). Additionally, an energy-dispersive X-ray spectrometer (EDS) featuring a silicon drift detector was utilized to analyze the chemical composition of the ULWC samples.

Compressive Strength and Density. After 28 days, compressive strength and density measurements EN 196-1 [44] were conducted. Three cubic specimens (100 × 100 × 100 mm) were tested to determine the average values for each parameter.

## 4. Results

### 4.1. Semi-Adiabatic Microcalorimeter

A semi-adiabatic microcalorimetry test was conducted to investigate the impact of waste glass powder content and high-frequency ultrasonic dispersion time on the hydration process of binders. Nine different mixtures were prepared, and their notation and detailed description are presented in Table 3. The reference mixture consisted solely of Portland cement (C100). In the other mixtures, varying portions of Portland cement were replaced with waste glass powder, and some binders were subjected to high-frequency ultrasonic dispersion.

In the reference mixture (C100), two key points were identified. The first, denoted as point 1, corresponds to the initial setting time, marked by the lowest temperature, during which most mixes exhibited satisfactory workability. The second, point 2, indicates the final setting time, marked by the highest temperature, where the mixture transitions from liquid to solid. These temperature peaks were distinctly observable in the mixtures where Portland cement was partially substituted with waste glass powder. However, the initial setting time became challenging when applying high-frequency ultrasonic dispersion.

During the research, it was observed that in the composition (C100), which did not contain any waste glass powder, the initial setting time occurred at 173 min, the final setting time was reached at 690 min, and the maximum temperature during hydration rose to 50.51 °C (Figure 5). When replacing 20% to 80% of Portland cement with waste glass powder, the initial setting time was delayed by approximately 7.55 h, from 173 min to 632 min, and the final setting time was delayed by about 12.53 h, from 690 min to 1444 min. Additionally, it was observed that substituting part of the Portland cement with waste glass powder decreased the temperature released during hydration. This suggests that in the early stages of hydration, waste glass powder acts as an inert filler and does not exhibit pozzolanic or cementitious properties.

To mitigate the negative impact of waste glass powder on the hydration process, high-frequency ultrasonic dispersion was employed. This technique induces cavitation in the dispersing medium, generating acoustic streams—shock waves that result from the explosion of cavitation bubbles. This reaction can momentarily raise the surface temperature of the material to 1000 °C and generate pressures up to 10,000 atm. This intense excitation is expected to disintegrate agglomerated particles and activate Portland cement and waste glass powder.

According to the obtained results (Figure 6 and Table 5), subjecting the binder composed of Portland cement and waste glass powder to high-frequency ultrasonic dispersion for periods ranging from 0 to 120 s accelerated the hydration process by approximately 8.37 h, reducing setting time from 988 min to 486 min. The observed slight increase in the final set temperature could be attributed to the higher initial temperature caused by the application of high-frequency ultrasonic dispersion (HFUS) rather than solely due to the increased reaction of Portland cement. The higher initial setting and solidification temperatures observed for C70/WG30 compared to C70/WG20 can be attributed to the increased concentration of alkali oxides (Na_2_O and K_2_O) in the waste glass powder. These oxides raise the pore solution’s alkalinity, accelerating early-stage hydration reactions and increasing heat generation during the initial setting phase.

However, it was also noted that longer ultrasonic dispersion led to higher initial mixture temperatures while still in the liquid phase. A slightly elevated initial mixture temperature will not significantly affect subsequent hydration processes. Still, a substantially increased initial temperature can accelerate the initial setting time, potentially making the mixture less workable. The synergy between waste glass powder and high-frequency ultrasonic dispersion lies in the latter’s ability to break down the glass particles and enhance their reactivity, thereby improving the overall performance of the binder and optimizing the hydration process.

### 4.2. XRD Analysis

Qualitative and quantitative XRD analyses were conducted to determine the phase composition of ultra-lightweight aluminum-based concrete. Figure 7 displays the XRD patterns of four hardened binders: C100, C70/WG30, C70/WG30 + 60 s of high-frequency ultrasonic (HFUS) dispersion, and C70/WG30 + 60 s of high-frequency ultrasonic (HFUS) dispersion + liquid glass.

Qualitative XRD analysis identified unreacted clinker phases in the reference composition (C100), such as C_3_S (*d*—0.8334; 0.5944 nm), C_2_S (*d*—0.5469; 0.4885 nm), C_3_A (*d*—0.4080; 0.3300 nm), and C_4_AF (*d*—0.7250; 0.5193 nm). Additionally, several hydration products were observed, including portlandite (*d*—0.4911; 0.3108 nm), ettringite (*d*—0.9720; 0.8850 nm), and various polymorphic forms of calcium carbonate such as calcite, vaterite, hemicarboaluminate, and monocarboaluminate. These calcium carbonate phases did not significantly affect the mechanical properties of the concrete and thus were not further investigated.

Results indicated that adding waste glass powder reduced the intensities of C_3_S and C_2_S. The application of HFUS further decreased clinker minerals, and the reduction was even more pronounced when combined with liquid glass. This reduction can be attributed to two factors: (1) the initial quantity of clinker phase decreased when Portland cement was partially substituted with waste glass powder, and (2) waste glass powder initially behaves as inert; however, in a highly alkaline environment and with the application of high-frequency ultrasonic dispersion, it accelerates clinker dissolution and promotes hydration by activating its pozzolanic properties at later stages. The combined methods of HFUS and liquid glass enhanced the hydration of Portland cement even further.

Quantitative XRD analysis via Rietveld refinement provided detailed insight into the mineral phase (Table 6). In the reference mixture (C100), the amorphous phase constituted 46.0%. Substituting 30% of Portland cement with waste glass powder (C70/WG30) increased the amorphous phase to 54.5%. This increase is attributed to the higher waste glass powder content and calcium silicate hydrate formation (C-S-H). During the hydration process, the tricalcium silicate (C_3_S) and dicalcium silicate (C_2_S) phases primarily contribute to the formation of calcium silicate hydrate (C-S-H), which is a crucial component affecting the mechanical properties of concrete. Samples subjected to HFUS and liquid glass treatments exhibited similar amorphous phase levels.

Waste glass powder proved effective in reducing the unreacted clinker phase. For example, substituting 30% of Portland cement with waste glass powder (C70/WG30) decreased the clinker phase (C_3_S + C_2_S) by 4.7%, from 19.3% to 14.6%. When high-frequency ultrasonic dispersion was applied for 60 s, the clinker phase (C_3_S + C_2_S) decreased by 6.5% from 19.3% to 12.8%. Combining HFUS with liquid glass resulted in an even more substantial decrease, with the clinker phase reducing by 10.8%, from 19.3% to 8.5%.

During the hydration process, the tricalcium silicate (C_3_S) and dicalcium silicate (C_2_S) phases primarily contribute to the formation of calcium silicate hydrate (C-S-H) and portlandite (CH). The reduction in these unreacted clinker phases highlights the enhanced reactivity due to the synergistic effects of waste glass powder and HFUS. Other unreacted clinker phases, which do not significantly affect the mechanical properties of hardened concrete, also showed a decrease.

The results indicate that substituting a portion of Portland cement with waste glass powder increased portlandite. Each additional method (HFUS and liquid glass) further increased portlandite content. Elevated portlandite levels favorably increased the pH value of the cement paste’s pore solution. A highly alkaline environment not only helps dissolve waste glass powder but also enables amorphous SiO_2_ from the glass powder to react with portlandite, forming low-basicity calcium silicate hydrate, which enhances the microstructure and mechanical properties of the concrete. Similar trends were observed with ettringite. However, ettringite tends to form in already existing voids and thus does not significantly impact mechanical properties. 

In summary, the synergistic application of waste glass powder, high-frequency ultrasonic dispersion, and liquid glass effectively enhances hydration, reducing unreacted clinker phases and increasing beneficial hydration products. These methods collectively improve the binder’s phase composition and microstructural characteristics, ultimately leading to better mechanical properties of concrete.

### 4.3. Mercury Intrusions Porosimeter Analysis

The mercury intrusion porosimetry test method was employed to evaluate the effects of waste glass powder, high-frequency ultrasonic dispersion, and liquid glass on pore size distribution. Pore sizes across all four tested compositions ranged from 4.8 nm to 351.8 µm, likely due to the device’s measurement limits. The total capillary porosity in the reference mixture (C100) was 58.49% (Figure 8). 

When 30% of Portland cement was replaced with waste glass powder (C70/WG30), porosity slightly increased by 1.68%, rising from 58.49% to 60.17%. Due to the impurities in waste glass powder, such as soil, food residues, or organic materials, additional porosity may be introduced during the mixing process. However, high-frequency ultrasonic dispersion effectively breaks down these air pockets, resulting in a denser structure than samples without ultrasonic activation. When combined with liquid glass activation, the structure becomes even more compact, as evidenced by the increased density in the final concrete specimens. However, applying high-frequency ultrasonic dispersion for 60 s (C70/WG30 + 60 s of HFUS dispersion) significantly reduced porosity by 7.45%, bringing it down to 51.04%. The most notable reduction was observed with the combination of ultrasonic dispersion and liquid glass (C70/WG30 + 60 s of HFUS dispersion + liquid glass), which lowered total porosity to 40.41%, an 18.08 decrease from the reference.

The pore size distribution for the reference mixture (C100) showed the highest concentration in the macro pore range (10–351.8 µm), with significant amounts in the capillary (0.03–10 µm) and gel pore (4.8 nm–0.03 µm) ranges (Figure 9). The substitution of 30% Portland cement with waste glass powder (C70/WG30) resulted in a marked reduction in macro pores while increasing the concentration of capillary pores. The gel pore concentration remained similar to the reference.

When high-frequency ultrasonic dispersion was applied (C70/WG30 + 60 s of HFUS dispersion), macro and capillary pore concentrations were further reduced, although gel pores slightly increased. Adding liquid glass to this treatment (C70/WG30 + 60 s of HFUS dispersion + liquid glass) slightly increased macro pore concentration, but capillary and gel pores were almost eliminated. After applying liquid glass treatment, the slight increase in macro porosity can be attributed to the absorption process under a negative vacuum. The rapid absorption of liquid glass into deeper structural layers may have created larger voids. A slower absorption process at lower negative pressure could mitigate this effect, and further optimization of this technology is necessary.

The combined use of waste glass powder, high-frequency ultrasonic dispersion, and liquid glass shows a synergistic effect on reducing total porosity and optimizing the pore size distribution within concrete. Waste glass powder alone slightly increases porosity, but its combination with ultrasonic dispersion and liquid glass significantly reduces it. Reduced macro porosity enhances concrete strength and decreases permeability, while lower capillary porosity improves compressive strength, reduces creep, and minimizes shrinkage. Though gel porosity minimizes compressive strength, it significantly affects shrinkage and creep.

Mercury intrusion porosimetry is less suitable for ultra-lightweight concrete despite its invasive nature. However, it remains a valuable tool for comparative studies. This research highlights the potential of integrating waste glass powder, high-frequency ultrasonic dispersion, and liquid glass to improve concrete properties by refining its pore structure. This approach offers a promising strategy for enhancing concrete performance through targeted porosity control.

### 4.4. Scanning Electron Microscopy

Figure 10 presents SEM micrographs of fractured surfaces in aluminum-based ultra-lightweight concrete for the following compositions: (a) C100, (b) C70/WG30, (c) C70/WG30 + 60 s of HFUS dispersion, and (d) C70/WG30 + 60 s of HFUS dispersion + liquid glass.

The reference composition (C100) reveals elongated, needle-like crystals characteristic of ettringite, confirmed by EDS analysis showing bulk regions of Ca-Al-S. Adjacent to the ettringite, areas indicative of calcium silicate hydrates (C-S-H) are observed, with EDS indicating a bulk area of Ca-Si. The fractured surface lacks plate-like crystals typically associated with portlandite and honeycomb-like structures indicative of C-S-H. Instead, regions rich in Ca are present, likely corresponding to portlandite or newly formed carbonates.

When 30% of Portland cement is substituted with waste glass powder, a more complex cement matrix (C70/WG30) is formed. The ettringite crystals become smaller and shorter, likely due to the increased pH caused by the high concentrations of Na_2_O and K_2_O in the waste glass powder. Other hydration products are less distinguishable due to their non-characteristic shapes, and the matrix is predominantly composed of calcium (Ca), silicon (Si), sodium (Na), and potassium (K).

Applying high-frequency ultrasonic dispersion for 60 s (C70/WG30 + 60 s of HFUS dispersion) produces an even denser cement paste. The surface is predominantly covered with more extended ettringite crystals, and no other hydration products are visibly identifiable.

Combining high-frequency ultrasonic dispersion with liquid glass (C70/WG30 + 60 s of HFUS dispersion + liquid glass) renders the cement matrix unrecognizable. The entire surface is coated with hardened liquid glass, obscuring ettringite, portlandite, and other hydration products. Table 7 presents an EDX spectrum analysis of all four fracture surfaces of hardened aluminum-based ultra-lightweight concrete.

The SEM micrographs demonstrate the progressive densification of the hardened cement paste with each treatment method. The reference composition (C100) exhibits typical ettringite and C-S-H formations. Introducing waste glass powder (C70/WG30) creates a denser matrix with smaller ettringite crystals due to elevated pH levels from the glass. High-frequency ultrasonic dispersion (C70/WG30 + 60 s of HFUS dispersion) further compacts the matrix, elongating the ettringite crystals. Combining ultrasonic dispersion and liquid glass (C70/WG30 + 60 s of HFUS dispersion + liquid glass) results in a highly dense matrix dominated by hardened liquid glass, obscuring individual hydration products.

These findings highlight the synergistic effect of waste glass powder, high-frequency ultrasonic dispersion, and liquid glass on the microstructure of aluminum-based ultra-lightweight concrete. The combined treatments lead to a denser and potentially more durable material, showcasing the potential for significant enhancements in concrete performance through targeted porosity management.

### 4.5. Density and Compressive Strength

The research demonstrates that while low substitution levels of waste glass powder have a relatively small impact on the compressive strength of aluminum-based ultra-lightweight concrete, higher substitution levels, particularly 60% and above, result in a more pronounced reduction in strength. In the reference mixture (C100), the compressive strength was measured at 3.1 MPa. When Portland cement was substituted up to 80% with waste glass powder, the compressive strength decreased by approximately 77%, from 3.1 MPa to 0.7 MPa (Figure 11). 

However, substituting 30% of Portland cement with waste glass powder (C70/WG30) and applying high-frequency ultrasonic (HFUS) dispersion for 30 to 90 s slightly increased compressive strength from 2.8 MPa to 3.1 MPa. Prolonging the HFUS dispersion to 120 s caused a minor decrease in compressive strength (Figure 12). This enhancement in compressive strength can be attributed to the synergistic effects of HFUS dispersion, which facilitates the deagglomeration and disintegration of binder particles, thereby accelerating the hydration processes of Portland cement and waste glass powder.

The study also found that neither the substitution of Portland cement with waste glass powder nor the application of HFUS dispersion significantly impacted the density of ultra-lightweight concrete (Figure 13). In all samples (C100, C70/WG30, and C70/WG30 + 60 s of HFUS dispersion), the density remained around 600 kg/m^3^. However, when samples were treated with liquid glass under a negative vacuum of −100 mbar for 30 min, the density increased by approximately 32%, from 602 kg/m^3^ to 799 kg/m^3^. This treatment also had a beneficial effect on compressive strength.

The compressive strength in the reference mixture (C100) was 3.1 MPa (Figure 14). Substituting 30% of Portland cement with waste glass powder (C70/WG30) reduced the strength by nearly 10%, from 3.1 MPa to 2.8 MPa. When HFUS dispersion was applied for 60 s to the C70/WG30 mixture (C70/WG30 + 60 s of HFUS dispersion), the compressive strength reverted to the reference value of 3.1 MPa. Subsequent liquid glass treatment on this composition significantly increased compressive strength by 87%, from 3.1 MPa to 5.8 MPa.

These findings suggest that substituting 30% of Portland cement with waste glass powder, HFUS dispersion, and liquid glass treatment markedly enhances the compressive strength of aluminum-based ultra-lightweight concrete. The synergistic interactions between these methods, particularly HFUS dispersion and liquid glass treatment, are crucial in achieving this improvement. While waste glass powder does not significantly enhance compressive strength when used alone, it is important to note that it significantly decreases compressive strength. Therefore, its impact must be considered in the context of overall material performance.

## 5. Discussion

The findings of this research highlight the complex interactions between waste glass powder, high-frequency ultrasonic (HFUS) dispersion, and liquid glass treatment on the hydration process and mechanical properties of aluminum-based ultra-lightweight concrete. The use of waste glass powder as a partial substitute for Portland cement shows a notable delay in the concrete mixtures’ initial and final setting times. This delay is evident as the initial setting time extended from 173 to 632 min, and the final setting time increased from 690 min to 1444 min when up to 80% of Portland cement was replaced by waste glass powder. The observed decrease in temperature during hydration and the delayed setting times suggest that waste glass powder behaves primarily as inert at early stages, lacking significant pozzolanic or cementitious activity.

Applying HFUS dispersion to the mixtures with 30% waste glass powder substitution introduced a positive shift in the hydration dynamics. HFUS dispersion effectively accelerated the hydration process, reducing the final setting time from 988 to 486 min over the dispersion period ranging from 0 to 120 s. This acceleration can be attributed to the cavitation effects induced by ultrasonic waves, which promote binder particles’ deagglomeration and enhance their reactivity. The dispersion not only improved the hydration kinetics but also resulted in a slight increase in compressive strength from 2.8 MPa to 3.1 MPa, counteracting the initial strength reduction caused by the inclusion of waste glass powder.

The synergistic effect of combining HFUS dispersion with liquid glass treatment was particularly noteworthy. This combination significantly enhanced both the density and compressive strength of the concrete. The density increased by approximately 32%, from 602 kg/m^3^ to 799 kg/m^3^, when the samples were treated with liquid glass under a negative vacuum. This treatment also led to an impressive 87% increase in compressive strength, from 3.1 MPa to 5.8 MPa. The densification of the concrete matrix and the improved mechanical properties can be attributed to the ability of liquid glass to fill and seal pores, thereby enhancing the overall integrity and durability of the material.

Mercury intrusion porosimetry analysis further elucidated these treatments’ impact on the concrete’s pore structure. The substitution of 30% Portland cement with waste glass powder resulted in a slight increase in total porosity. However, the application of HFUS dispersion significantly reduced porosity, with the most substantial reduction observed when combined with liquid glass treatment. This reduction in porosity, particularly in the macro and capillary pore ranges, contributes to the enhanced mechanical properties and reduces the permeability of the concrete. The lower porosity levels correlate with improved compressive strength, reduced creep, and minimized shrinkage.

Scanning electron microscopy (SEM) provided microstructural insights, revealing progressive densification of the cement matrix with each applied treatment. The reference composition (C100) displayed typical ettringite and calcium silicate hydrate (C-S-H) formations. Incorporating waste glass powder leads to a more refined matrix, but it is important to note that the overall porosity increases due to impurities in the waste glass. In contrast, applying high-frequency ultrasonic dispersion and using liquid glass significantly reduce porosity, resulting in a more optimized microstructure. The application of HFUS dispersion further compacted the matrix, elongating ettringite crystals. The combination of HFUS dispersion and liquid glass treatment resulted in a highly dense matrix dominated by hardened liquid glass, obscuring individual hydration products and indicating a significant improvement in the material’s microstructural characteristics.

The research demonstrated that while waste glass powder alone does not significantly enhance ultra-lightweight concrete’s compressive strength or hydration kinetics, its combination with HFUS dispersion and liquid glass treatment leads to substantial improvements. The synergistic effects of these methods optimize the hydration process, reduce porosity, and enhance the microstructure, resulting in a concrete material with superior mechanical properties and durability. These findings provide valuable insights into the potential applications of waste glass powder and advanced treatment methods in developing high-performance, sustainable concrete materials.

The hydration products formed under different treatment methods significantly influence the microstructure and properties of aluminum-based ultra-lightweight concrete. Under high-frequency ultrasonic dispersion, calcium silicate hydrate (C-S-H) formation is promoted, leading to a more uniform particle distribution and potentially enhanced mechanical properties. In contrast, incorporating waste glass powder at varying percentages has affected the type and quantity of hydration products. Specifically, higher proportions of waste glass may result in increased porosity, as the glass particles can hinder the formation of denser C-S-H structures. Furthermore, the addition of liquid glass activation appears to enhance the development of C-S-H, contributing to improved bonding and reduced porosity. This interplay of hydration products highlights the importance of treatment methods in optimizing the performance characteristics of the concrete.

## 6. Conclusions

Substituting up to 80% of Portland cement with waste glass powder significantly delays the hydration process, reducing compressive strength by approximately 77%, which indicates the inert nature of waste glass powder during early hydration stages.The application of high-frequency ultrasonic (HFUS) dispersion to mixtures with 30% waste glass powder replacement results in a slight increase in compressive strength, suggesting that HFUS dispersion accelerates hydration by effectively deagglomerating and disintegrating binder particles.Combining HFUS dispersion with liquid glass treatment markedly improves compressive strength and density, demonstrating the synergistic benefits of these methods in optimizing binder reactivity and enhancing concrete microstructure.HFUS dispersion and liquid glass treatment significantly reduce total porosity and refine pore size distribution, improving the mechanical properties of ultra-lightweight concrete.Scanning electron microscopy analysis shows progressive densification of the cement matrix with each application, with the HFUS and liquid glass combination resulting in a highly compact material. This indicates a significant enhancement in the concrete’s microstructural characteristics.

## Figures and Tables

**Figure 1 materials-17-05430-f001:**
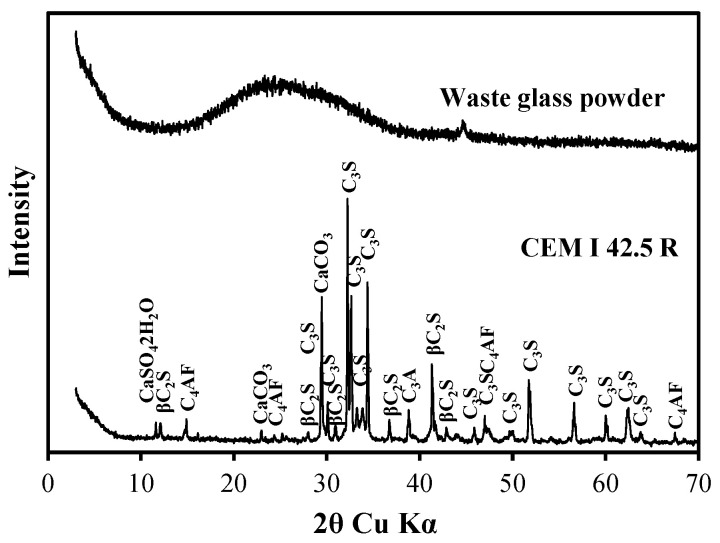
XRD patterns of unhydrated Portland cement (CEM I 42.5 R) and waste glass powder.

**Figure 2 materials-17-05430-f002:**
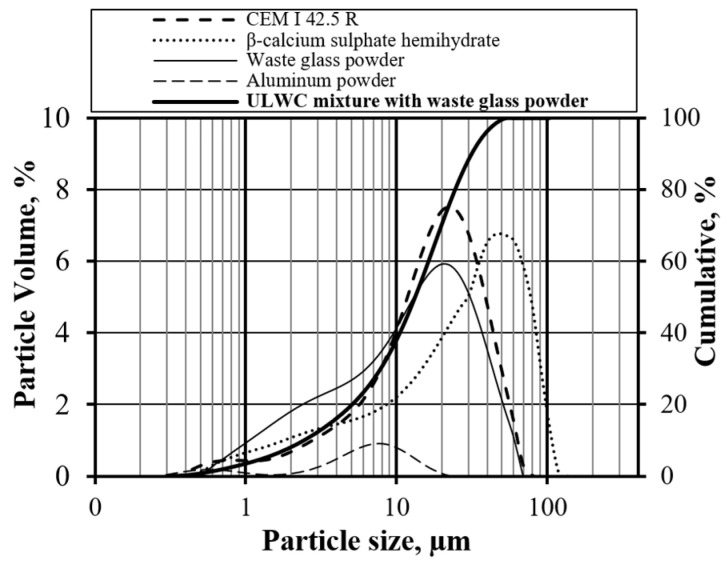
Particle size distribution of Portland cement (CEM I 42.5 R), β-calcium sulfate hemihydrate, waste glass powder, aluminum powder, and aluminum-based ultra-lightweight concrete mix.

**Figure 3 materials-17-05430-f003:**
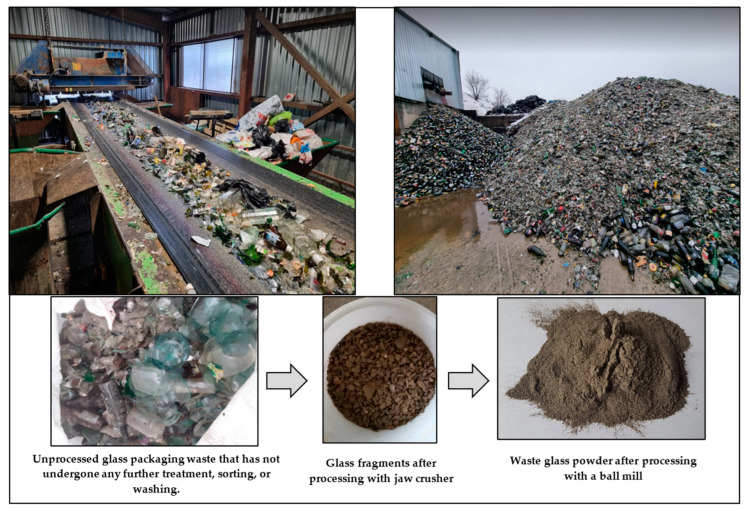
The conversion process of waste glass shards into waste glass powder.

**Figure 4 materials-17-05430-f004:**
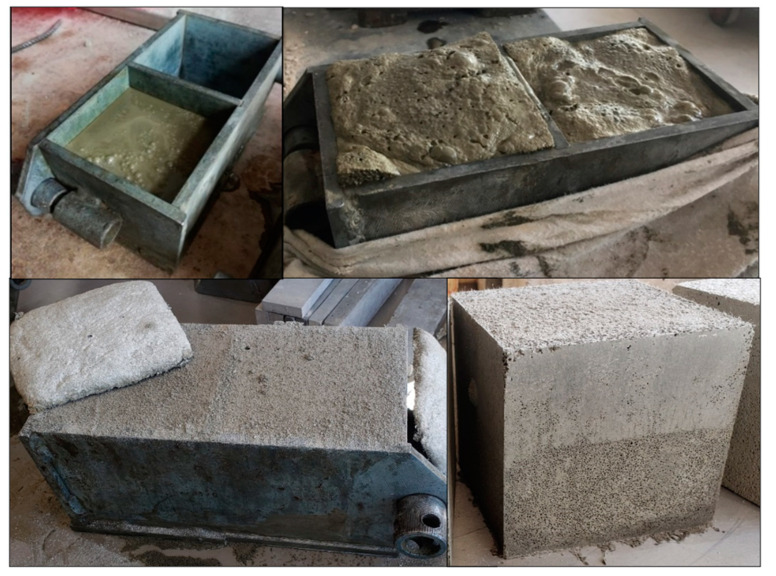
Transformation of the slurry mixture into aluminum-based ultra-lightweight concrete with waste glass powder.

**Figure 5 materials-17-05430-f005:**
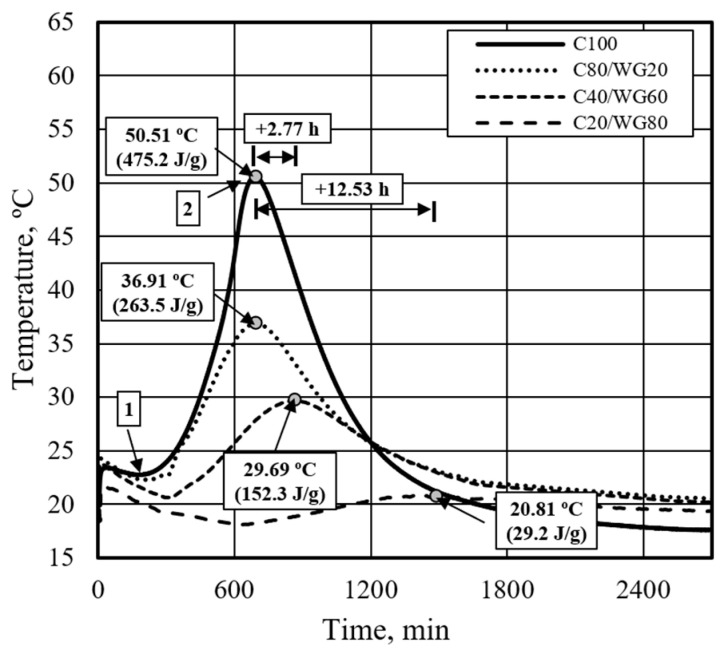
Effect of waste glass powder on the hydration process of Portland Cement.

**Figure 6 materials-17-05430-f006:**
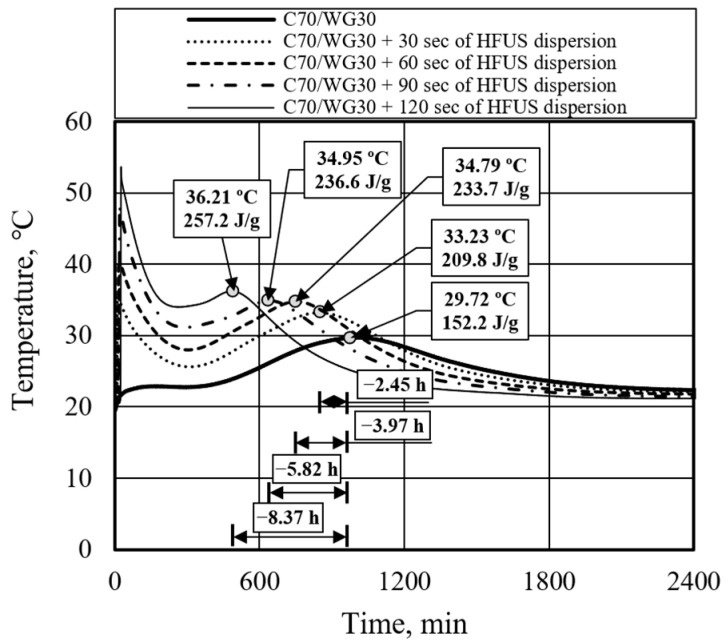
The impact of high-frequency ultrasonic dispersion duration on the binder’s hydration process.

**Figure 7 materials-17-05430-f007:**
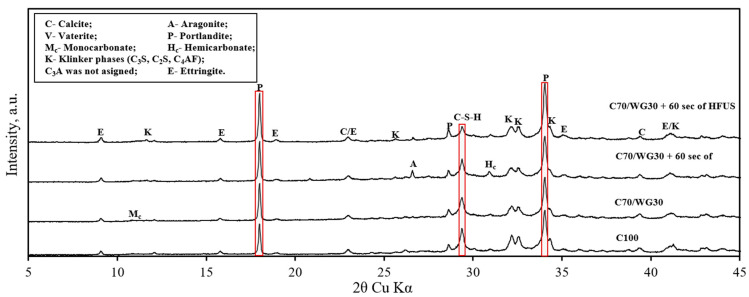
Effect of waste glass powder, high-frequency ultrasonic dispersion, and liquid glass on the XRD patterns of ULWC (W/B = 0.71).

**Figure 8 materials-17-05430-f008:**
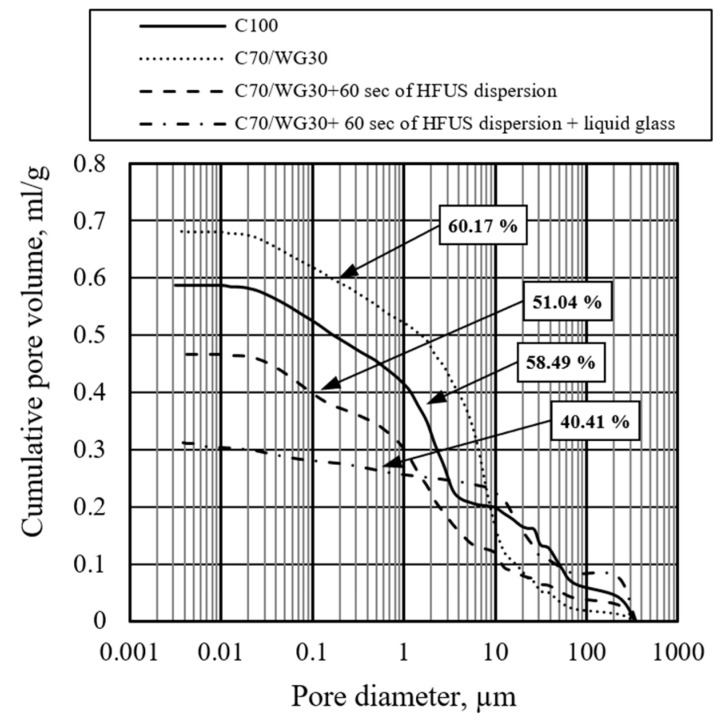
Effect of waste glass powder, ultrasonic dispersion, and liquid glass on the cumulative pore volume.

**Figure 9 materials-17-05430-f009:**
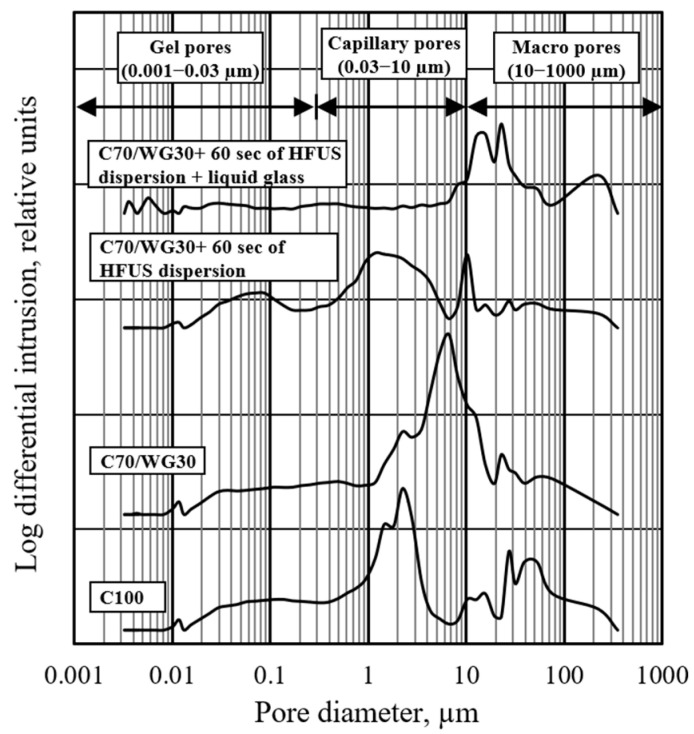
Effect of waste glass powder, ultrasonic dispersion, and liquid glass on the log differential intrusion.

**Figure 10 materials-17-05430-f010:**
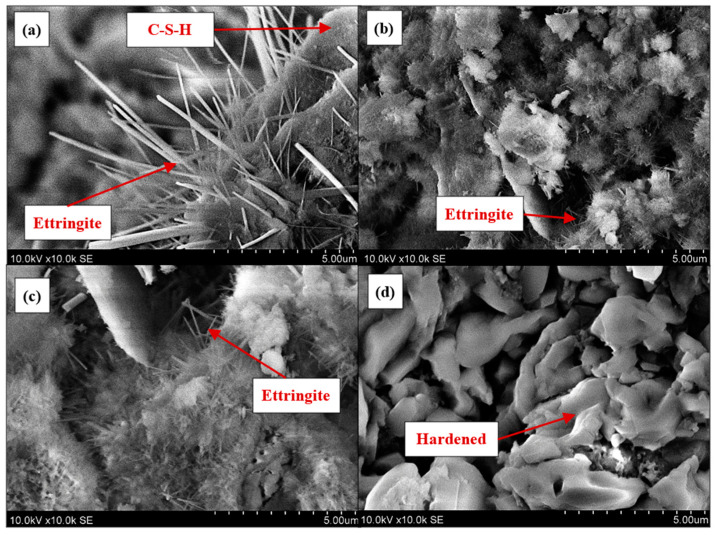
SEM micrographs of fractured surfaces of aluminum-based ultra-lightweight concrete: (**a**) C100, (**b**) C70/WG30, (**c**) C70/WG30 + 60 s of HFUS, (**d**) C70/WG30 + 60 s of HFUS dispersion + liquid glass.

**Figure 11 materials-17-05430-f011:**
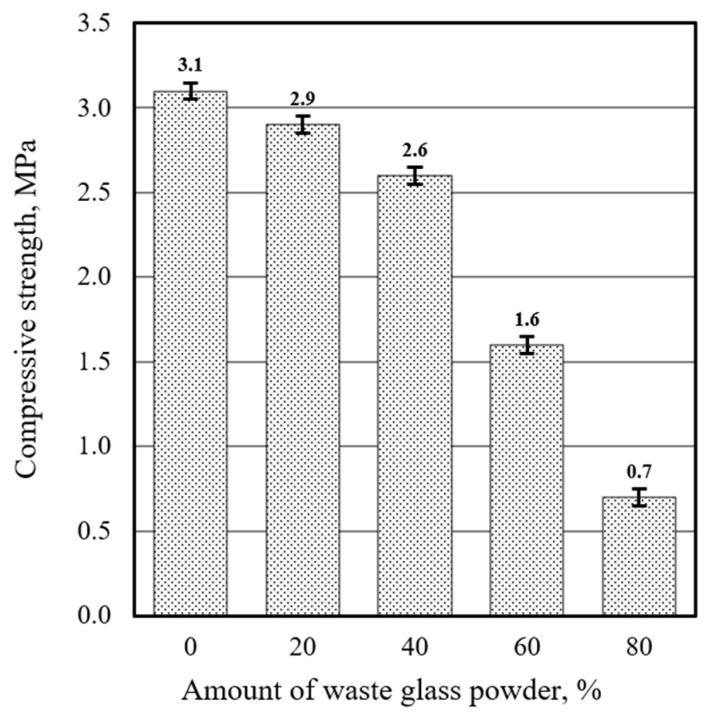
Effect of Portland cement replacement ratio to waste glass powder on the compressive strength (after 28 days).

**Figure 12 materials-17-05430-f012:**
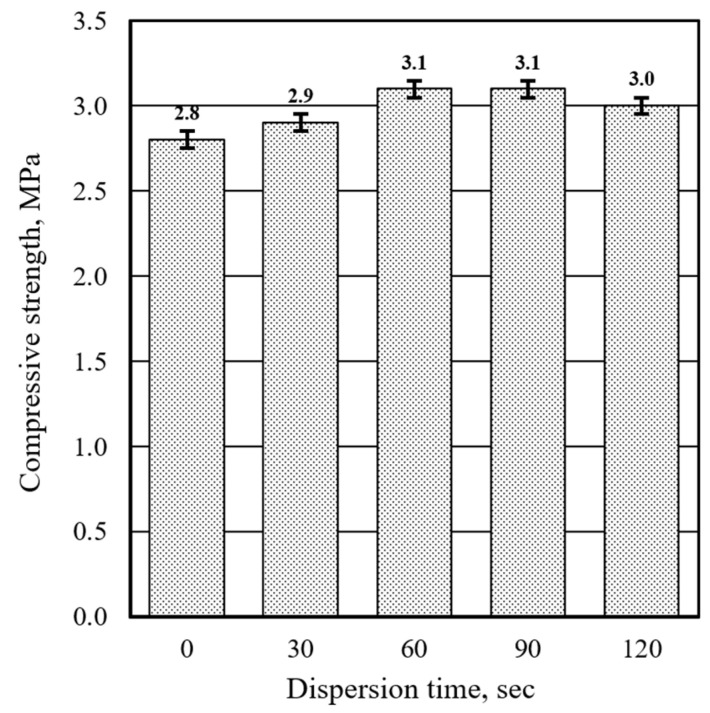
Effect of high-frequency ultrasonic dispersion time on the compressive strength (C70/WG30, after 28 days).

**Figure 13 materials-17-05430-f013:**
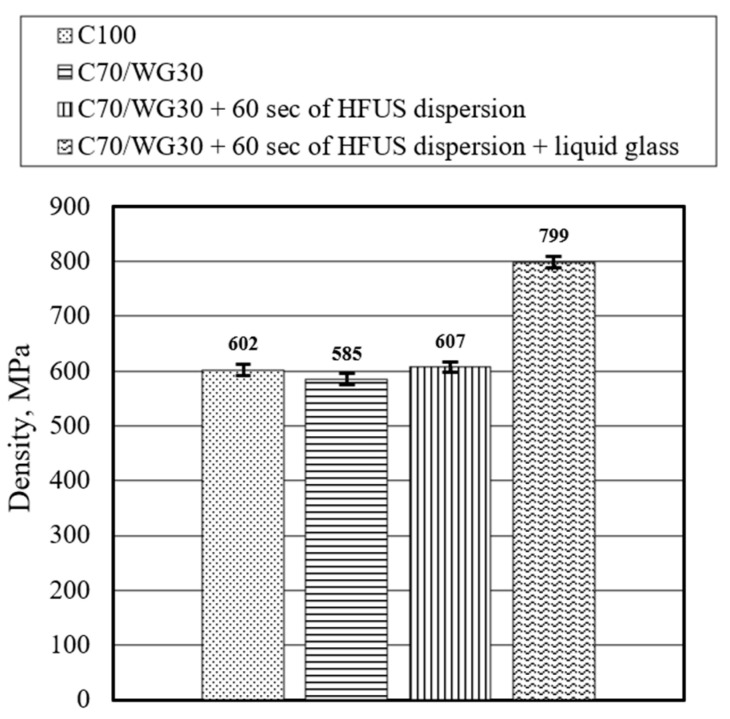
Effect of waste glass powder, high-frequency ultrasonic dispersion, and liquid glass on the density of ULWC (after 28 days).

**Figure 14 materials-17-05430-f014:**
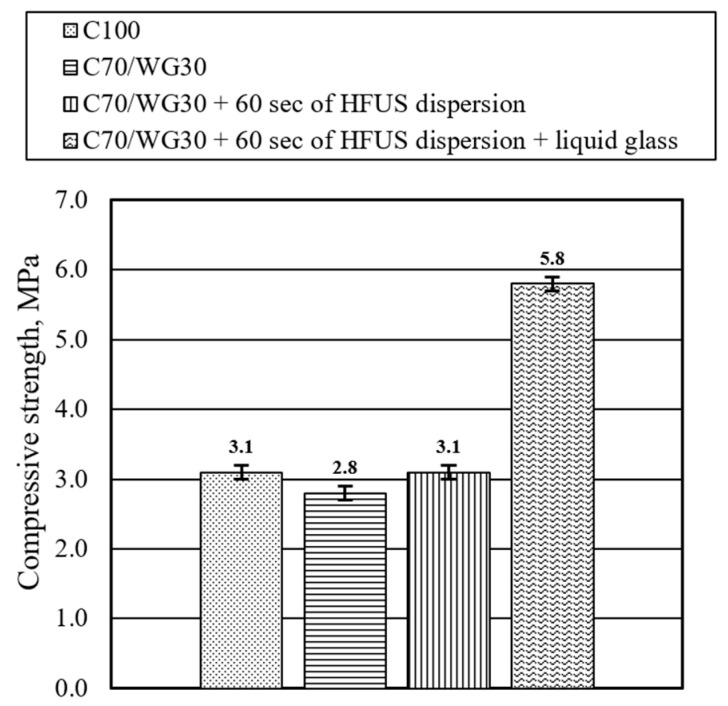
Effect of waste glass powder, high-frequency ultrasonic dispersion, and liquid glass on the compressive strength of ULWC (after 28 days).

**Table 1 materials-17-05430-t001:** Chemical composition of Portland cement (CEM I 42.5 R) and waste glass powder.

Chemical Composition	Quantity %	
	CEM I 42.5 R	Waste Glass Powder
SiO_2_	21.11	54.80
TiO_2_	0.23	-
Al_2_O_3_	3.41	3.45
Fe_2_O_3_	5.24	2.08
MnO	0.06	-
MgO	0.75	2.41
CaO	65.41	15.23
SO_3_	1.94	1.25
Na_2_O	0.20	8.25
K_2_O	0.31	1.95
P_2_O_5_	0.29	-
Na2Oeq.	0.45	9.49
Loss of ignition	0.58	1.05

**Table 2 materials-17-05430-t002:** Mixing procedure of aluminum-based ultra-lightweight concrete with waste glass powder.

Duration, s	Procedure
60	Homogenization of cement and waste glass powder with a high-shear mixer (500 rpm).
60	Gradually add premeasured water to the homogenized slurry and mix with a high-shear mixer at 700 rpm.
0–120	Another crucial step is applying a high-frequency ultrasonic dispenser to the homogenized slurry. This procedure aids particle deagglomeration, enhances particle distribution, and promotes binder hydration.
15	Dilute the sodium hydroxide in a separate container (deduct water from the total water required for the mixture). Pour the sodium hydroxide solution into the slurry mix. Continue to mix with a high-shear mixer at 700 rpm.
15	Carefully add gypsum to the slurry mix. Reduce high-shear mixing speed to 600 rpm.
15	Carefully add the aluminum powder to the slurry mix. Keep high-shear mixing rotation speed at 600 rpm.
60	Gradually add the polypropylene fibers to the slurry mix. Increase the rotation speed to 800 rpm.

**Table 3 materials-17-05430-t003:** Notation and description of the binder composition.

No.	Binder Composition	Description
Influence of Waste Glass Powder		
1	C100	The composition consists of 100% Portland cement CEM I 42.5 R.
2	C80/WG20	The composition comprises 80% Portland cement CEM I 42.5 R and 20% waste glass powder.
3	C40/WG60	The composition comprises 40% Portland cement CEM I 42.5 R and 60% waste glass powder.
4	C20/WG80	The composition comprises 20% Portland cement CEM I 42.5 R and 80% waste glass powder.
Influence of high-frequency ultrasonic dispenser time		
5	C70/WG30	The composition comprises 70% Portland cement CEM I 42.5 R and 30% waste glass powder.
6	C70/WG30 + 30 s of HFUS dispersion	The composition comprises 70% Portland cement CEM I 42.5 R and 30% waste glass powder. The binder is further dispersed for 30 s using a high-frequency ultrasonic disseminator.
7	C70/WG30 + 60 s of HFUS dispersion	The composition comprises 70% Portland cement CEM I 42.5 R and 30% waste glass powder. The binder is further dispersed for 60 s using a high-frequency ultrasonic disseminator.
8	C70/WG30 + 90 s of HFUS dispersion	The composition comprises 70% Portland cement CEM I 42.5 R and 30% waste glass powder. The binder is further dispersed for 90 s using a high-frequency ultrasonic disseminator.
9	C70/WG30 + 120 s of HFUS dispersion	The composition comprises 70% Portland cement CEM I 42.5 R and 30% waste glass powder. The binder is further dispersed for 120 s using a high-frequency ultrasonic disseminator.
10	C70/WG30 + 60 s of HFUS dispersion + liquid glass	The composition comprises 70% Portland cement CEM I 42.5 R and 30% waste glass powder. The binder is further dispersed for 120 s using a high-frequency ultrasonic disseminator. The next day, after demolding, ULWC cubes were submerged in liquid glass and kept under a negative vacuum for 30 min.

**Table 4 materials-17-05430-t004:** Compositions of aluminum-based ultra-lightweight concrete.

Components	Composition			
	C100	C70/WG30	C70/WG30 + 60 s of HFUS Dispersion	C70/WG30 + 60 s of HFUS Dispersion + Liquid Glass
W/B	0.71	0.71	0.71	0.71
Water, kg/m^3^	462			
Cement, kg/m^3^	654	458		
Waste glass powder, kg/m^3^	-	196		
Gypsum, kg/m^3^	1.80			
NaOH, kg/m^3^	2.70			
Aluminum powder, kg/m^3^	0.68			
Polypropylene fiber, kg/m^3^	1.83			
Liquid glass, kg/m^3^	-	-	-	~(180–200)

**Table 5 materials-17-05430-t005:** Effect of waste glass powder and high-frequency ultrasonic dispersion time on the hydration process of Portland cement.

Composition	Initial Temperature, °C	Initial Setting Temperature, °C	Initial Setting Time, min	Final Setting Temperature, °C	Final Setting Time, min	Change in Final Setting, min	Heat of Hydration, J/g
Influence of Waste Glass Powder							
C100	19.59	22.75	173	50.51	690	0	475.2
C80/WG20	19.56	22.28	228	36.88	692	2	263.5
C40/WG60	19.58	20.65	315	29.65	856	166	152.3
C20/WG80	19.32	18.08	632	20.75	1444	752	29.2
Influence of high-frequency ultrasonic dispenser time							
C70/WG30	19.76	22.70	175	29.72	988	0	152.2
C70/WG30 + 30 s of HFUS dispersion	34.29	-	-	33.23	841	−147	209.8
C70/WG30 + 60 s of HFUS dispersion	39.47	-	-	34.79	750	−238	233.7
C70/WG30 + 90 s of HFUS dispersion	48.2	-	-	34.95	639	−349	236.6
C70/WG30 + 120 s of HFUS dispersion	53.52	-	-	36.21	486	−502	257.2

**Table 6 materials-17-05430-t006:** Effect of waste glass powder, ultrasonic dispersion, and liquid glass on the mineralogical composition of aluminum-based ultra-lightweight concrete.

Phase	C100	C70/WG30	C70/WG30 + 60 s of HFUS Dispersion	C70/WG30 + 60 s of HFUS Dispersion + Liquid Glass
Amorphous	46.0	54.5	55.7	55.4
C_3_S monoclinic	9.1	6.5	5.0	3.8
C_2_S beta	10.2	8.1	7.8	4.7
C_3_A cubic	0.5	0.2	0.1	-
C_4_AF Colville	7.5	6.8	5.1	3.9
Calcite	8.3	6.7	4.4	3.2
Aragonite	4.5	1.8	1.6	-
Portlandite	7.3	8.3	11.0	15.4
Ettringite	4.8	3.7	4.9	8.8
Hemicarboaluminate	0.6	1.4	1.1	0.9
Monocarbonate	1.1	2.1	3.2	3.9

**Table 7 materials-17-05430-t007:** EDX spectrum analysis of a fractured surface of hardened aluminum-based ultra-lightweight concrete.

Chemical Element	C100	C70/WG30	C70/WG30 + 60 s of HFUS Dispersion	C70/WG30 + 60 s of HFUS Dispersion + Liquid Glass
O	39.61	42.41	38.70	45.67
Si	13.83	6.83	9.78	23.39
C	1.90	1.70	0.62	0.16
Na	2.16	0.39	0.56	10.33
Ca	36.30	42.18	43.33	17.87
Mg	1.11	1.19	1.23	0.22
Al	1.49	1.03	1.65	0.56
K	0.85	0.87	0.72	0.45
S	1.15	0.37	1.52	0.42
Fe	1.60	3.03	1.89	0.93

## Data Availability

The original contributions presented in the study are included in the article. Further inquiries can be directed to the corresponding author.

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
