# Peer review of "Synergistic Effects of Waste Glass Powder, High-Frequency Ultrasonic Dispersion, and Liquid Glass Treatment on the Properties of Aluminum-Based Ultra-Lightweight Concrete"

_materials, 2024, doi:10.3390/ma17225430_

Round 1

Reviewer 1 Report

Comments and Suggestions for Authors

please find the attachment

Author Response

I am very thankful to the reviewers for their deep and thorough review. I have revised my present research paper considering their valuable suggestions and comments. I hope my revision has improved the paper to their satisfaction. The number-wise answers to their specific comments are as follows.

Reviewer 2 Report

Comments and Suggestions for Authors

The overall topic is interesting and the research study is well designed.  The paper is well written and easy to follow.  I have a few suggestions/questions to improve the paper.

- It may just be my version, but in Figure 3 the captions appear cutoff

- Line 173 - Somewhere around here it would help to add the size of the samples cast and if the samples were cut or otherwise modified for the different tests.  You do list sizes for MIP and compressive strength down below, but were the same samples used for XRD and SEM?

- Line 245-246, I don't think you can say that the final set time increased due to more reacted portland cement.  It seems to me that the small increase in final set temperature you saw could also be attributed to the higher initial temperature caused by the HFUS.

- Line 371 - I don't know that you can say it the C70/WG30 creates a denser matrix as it has a higher void percentage than the C100.  Maybe a more refined matrix is what you are trying to indicate?

- Line 383 - I don't think you can say glass powder has a negligible effect on compressive strength as it significantly decreases the strength.  When you add in HFUS it brings some strength back but this statement is misleading.

- Line 386-387 - Similar comment to above. Saying that waste glass powder alone does not significantly enhance compressive strength is true but misleading as it significantly decreases compressive strength when used alone.

- Line 444 - As above, you are saying waste glass leads to a denser matrix, but void ratio is larger so this is not true.

- It may be outside the scope of your work, but do you know the effect of HFUS on 100% cement material?  If this is something you have done or if documentation exists in the literature it would be interesting information to include.

- A general formatting comment is that the figures don't seem to align well with the references in the text.  For some figures (Figure 11 for example) it comes 3 paragraphs before the figure is mentioned in the text.  It would be beneficial to have the figures soon after they are referenced.

Author Response

(The authors gave the same response as above.)

Round 2

Reviewer 1 Report

Comments and Suggestions for Authors

what is “CSH”? authors are suggested to check the basic idea of cement chemistry or compare previous works. it is "C-S-H", rather than "CSH". Please resive the manuscript again according to reviewers' comments.

Comments on the Quality of English Language

none

Author Response

Reviewer’s comment:
“What is ‘CSH’? Authors are suggested to check the basic idea of cement chemistry or compare previous works. It is ‘C-S-H,’ rather than ‘CSH’. Please revise the manuscript again according to reviewers' comments.”

Response to the reviewer:
Thank you for your comment regarding the notation of "C-S-H." We agree that the correct term in cement chemistry should indeed be "C-S-H" (calcium silicate hydrate). We have carefully reviewed the manuscript and replaced all instances of "CSH" with "C-S-H" in accordance with standard cement chemistry notation. Additionally, we have checked similar terms to ensure consistent usage throughout the manuscript.

The corrections have been highlighted in green for easier reference.